# LEARNING A METACOGNITION FOR OBJECT DETECTION

## ABSTRACT

In contrast to object recognition models, humans do not blindly trust their perception when building representations of the world, instead recruiting metacognition to detect percepts that are unreliable or false, such as when we realize that we mistook one object for another. We propose METAGEN, an unsupervised model that enhances object recognition models through a metacognition. Given noisy output from an object-detection model, METAGEN learns a meta-representation of how its perceptual system works and uses it to infer the objects in the world responsible for the detections. METAGEN achieves this by conditioning its inference on basic principles of objects that even human infants understand (known as Spelke principles: object permanence, cohesion, and spatiotemporal continuity). We test METAGEN on a variety of state-of-the-art object detection neural networks. We find that METAGEN quickly learns an accurate metacognitive representation of the neural network, and that this improves detection accuracy by filling in objects that the detection model missed and removing hallucinated objects. This approach enables generalization to out-of-sample data and outperforms comparison models that lack a metacognition.

## 1 INTRODUCTION

Learning accurate representations of the world is critical for prediction, inference, and planning in complex environments (Lake et al., 2017). In human vision, these representations are generated by a perceptual system that transforms light entering the retina into representations of the physical space and the objects in it (Kar & DiCarlo, 2021; Güçlü & van Gerven, 2017). While human perception is generally robust and reliable, it nonetheless suffers from errors, such as when we fail to detect an object in a crowded visual scene, when we confuse one object for another, or when we experience a visual illusion. Critically, in these situations, people have a *metacognition* that helps them recognize that their visual system has failed them and adjust their reasoning accordingly.

End-to-end object recognition models have revolutionized computer vision in the past decade (Krizhevsky et al., 2012; Redmon et al., 2016; Zou et al., 2019), but they face a similar challenge: how can we identify when a non-existent object is incorrectly detected (hallucinations), or when a real object is not reliably recognized (misses)? Traditional approaches to this problem have focused on increasing the training data (Lin et al., 2014; Deng et al., 2009) or improving the model's architecture (Carion et al., 2020; Liu et al., 2021). While these methods are generally successful, the possibility of misses and hallucinations remains—particularly when confronted with out-of-sample data—and a foolproof visual model must be able to identify when this occurs. We propose that augmenting object recognition models with a metacognition—an external meta-representation of their own computational processes—can allow these systems to monitor their percepts and flexibly decide when to trust or doubt proposed representations, much like humans do.

In this paper, we propose a formulation of metacognition that can be learned online, requiring no human annotation or feedback. Given a pre-trained object detection model and a data stream from videos or scene viewpoints, our approach is to perform a joint inference over the objects generating the percepts, and a meta-representation of the system's performance. Our model, *METAGEN*, achieves this by representing its metacognition as a generative model that captures the joint distribution between objects in a scene and the model's pattern of detections. Critically, we make this problem tractable by drawing on an insight from cognitive science. Infants come into the world

equipped with a basic form of 'core knowledge' or 'start-up sofware' (Lake et al., 2017; Carey, 2009). These priors constrain the representations of world states that humans consider possible and support drawing richer inferences with less data. Specifically, we focus on early object constraints known as *Spelke principles* (Spelke & Kinzler, 2007; Smith et al., 2019): objects persist in time, they cannot occupy the same position as another object, and they move continuously in space. Conditioning inference on these basic principles enables METAGEN to learn a metacognitive representation of the object detection model by analyzing and resolving patterns of percepts that violate the Spelke principles of objects.

We evaluate METAGEN in a variety of ways and against several comparison models, focused on performance on out-of-sample data. Using scenes rendered in in the ThreeDWorld (TDW) virtual environment (Gan et al., 2020), we first examine METAGEN's capacity to learn accurate metacognition online for a variety of state-of-the-art object detection networks that span different dominant paradigms (single-stage, two-stage, and transformer models). We find that, during learning, META-GEN already produces marked improvements over key comparison models: the object detection model without a metacognition and the object detection models with a confidence threshold fit to the data. We then evaluate METAGEN after learning by conditioning on its learned metacognition and testing its accuracy on a new set of data. We find that METAGEN also outperforms comparison models on a set of novel scenes.

Overall, our work makes two main contributions. First, we propose a novel approach to improving object detection models through a metacognition that helps monitor a model's performance. This distinction between perceptual processing systems and high-level cognitive systems mirrors the architecture of the human mind (Firestone & Scholl, 2016) where both processes work jointly to build accurate representations of the world. Second, we present METAGEN, a model that learns a metacognition for object detection models. We show that METAGEN can learn an accurate representation of object detection models without feedback and leads to rapid improvements in the system's accuracy.

## 2 RELATED WORK

**Metacognition in AI.** Previous work has argued for the importance of metacognition for machine learning and AI (Cox, 2005). Models that use metacognition during learning have shown promise for improving classification accuracy (Babu & Suresh, 2012; Subramanian et al., 2013). This work has focused on engineering an inflexible metacognition to guide learning. In this paper, we focus on a complimentary problem: learning a metacognition without feedback that helps monitor a neural network's detections.

**Computational cognitive science.** Our core idea—learning a metacognitive representation of a perceptual system—is inspired by research in cognitive science showing that human reasoning is structured around mental models of the physical world (Ullman et al., 2017; Battaglia et al., 2013), of the social world (Jara-Ettinger et al., 2019; Baker et al., 2017), and of ourselves (Gopnik, 1993; Nisbett & Wilson, 1977). Our work is also related to computational models of human core knowledge (Smith et al., 2019; Kemp & Xu, 2009). The difference is that our work uses object principles to learn a metacognition, whereas past work has worked on modeling object principles themselves.

**Uncertainty-aware AI.** The spirit of our work relates more closely to uncertainty-aware AI. This work focuses on building end-to-end systems that express uncertainty in their inferences (Sensoy et al., 2018; Kaplan et al., 2018; Ivanovska et al., 2015). Our work focuses instead on how to learn a model of uncertainty over a pre-trained model. These two approaches complement each other. In humans, metacognitive uncertainty supplements the intrinsic uncertainty in visual perception.

**Meta-Learning.** A major challenge for machine learning algorithms is generalizing to new datasets and tasks that they were not trained on. Recent and exciting work in machine learning has explored the possibility of meta-learning (Hospedales et al., 2020), which involves applying a model trained on one task to a new setting (Finn et al., 2017). At first glance, our work may appear to fall within the field of meta-learning. However, in contrast to meta-learning, METAGEN does not improve the first-level learning algorithm (the neural network for object detection). Instead, METAGEN functions at a level above learning algorithm, taking the outputs of the learning algorithm as unchangeable inputs.

# 3 METAGEN

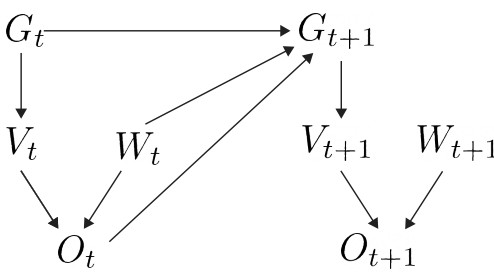

Figure 1: Schematic depicting the forward generative model. $G_t$ is the prior over metacognition ($V_t$) at time t; $W_t$ is the world state; and $O_t$ are the observations that are generated. $W_t$, $O_t$, and $G_t$ collectively influence $G_{t+1}$, the prior over $V_{t_1}$.

In this section we present our METAGEN model, applied to the problem of inferring what objects are in a 3D scene, given a video from a moving camera. METAGEN captures the relationship between the objects in a scene and the detections obtained by its perceptual module (i.e., the output from an object recognition model) through a hierarchical generative model with two levels (Figure 1). The *lower* level captures the joint distribution between world states and object detections, as determined by a metacognitive representation $V$ that summarizes its perception module's performance. The *upper* level describes the dynamics of the prior over V, capturing the idea that a metacognitive representation can—and should—change with experience.

Given a collection of observations $\vec{o}$ obtained from a scene, our approach is to perform a joint inference over the unobservable world state $W$ and the metacognitive representation $V$, conditioned on Spelke principles of objects. We begin by presenting the generative model (3.1) and then turn to how we perform joint inference over the world state and the system's metacognition (3.2).

## 3.1 GENERATIVE MODEL

### 3.1.1 WORLD-STATE REPRESENTATION AND SPELKE PRINCIPLES

Let $\mathbb{W}$ be the space of possible world states, where $W \in \mathbb{W}$ consists of a collection of objects, each associated with a category and with a 3D position in space. Throughout, we use $W_t$ to refer the world state from the $t$-th scene encountered by METAGEN, and $\vec{W} = \{W_0, \ldots, W_T\}$ to refer to the complete sequence of world states.

In our approach, Spelke principles constrain possible world states. Specifically, the assumption of object permanence implies that each world state has a fixed set of objects that does not change. Next, the assumption that two objects cannot occupy the same physical position is implemented as a prior over the locations of the objects in a world state. The prior over the location of an object is a uniform over 3-D space, except near another object, where the probability of placement decreases to $0$. Finally, the assumption of spatio-temporal continuity implies that any moving object must exhibit a continuous trajectory. Our Experimental task focuses on inferring permanent fixtures in a room, so we therefore do not implement this constraint, although doing so is a solved problem (Smith et al., 2019; Kemp & Xu, 2009). Implementation details are available in the Appendix.

### 3.1.2 PERCEPTUAL MODULE

The goal of METAGEN is to build a metacognitive representation of any object detection model that can transform images into object detections, each associated with a category and a position in the 2D image. Although METAGEN does not have access to ground truth ($W$), we assume access to the camera's state $cs$, which determines its position in space and orientation. This helps with inferring and tracking the location of different objects (as this is not the main focus of our work). We additionally assume access to video boundaries, enabling METAGEN to reset object representations across scenes. Critically, the perceptual models functionally serves as a black box, and METAGEN does not have any access to the internal state of the model, knowledge about its performance, or any feedback about the final inferences.

Formally, let $O_t = \vec{o}$ be the collection of observations associated with the $t$-th scene. Each observation $o_{cs} \in O_t$ is a percept associated with camera state $cs$ (from the complete camera trajectory $cj$),

and consists of a set of point detections obtained from the object recognition model, each containing the object's category and 2-D location on an image: $d = (c, x, y)$.

### 3.1.3 METACOGNITIVE REPRESENTATION

The metacognitive representation of the perceptual model, $V$, consists of two probability distributions per object category. The first distribution captures the model's propensity to hallucinate objects, represented through a Poisson distribution with rate $\lambda_c$ for object category $c$. The second distribution captures the model's propensity to correctly detect an object that's in view. We represent this through a Geometric distribution with rate $p_c$, which helps us account for the fact that object models can sometimes detect the same object multiple times. Under this formulation, the model's miss rate for an object in category $c$ is $1 - p_c$.

Because each distribution is captured by a single parameter, the perceptual module's metacognition $V$ can then be represented as a $|C|$-by-2 matrix storing each category's hallucation rate $\lambda_c$ and miss rate $1 - p_c$. This defines a generative model such that $V(O, W, cs)$ describes the probability that the visual system would produce percept $O$ when processing world state $W \in \mathbb{W}$ with the camera state $cs$. This is calculated by processing every object in $W$ that is visible from camera state $cs$, through the metacognitive representation captured in $V$.

### 3.1.4 METACOGNITIVE DYNAMICS

The generative model described so far captures how METAGEN learns a metarepresentation of its perceptual module's performance. However, an object detection model's propensity to hallucinate or miss objects can vary across scenes, and leaving flexibility in $V$ to be adjusted is a desirable property. At the same time, experience in a previous scene includes critical information about the perceptual module that should inform expectations about its performance in a new scene. Our generative model therefore includes an evolving kernel $G_t$ (Figure 1), capturing changing priors over $V$.

When $t = 0$, each category's initial hallucination rate $\lambda_c$ is initialized from a Gamma distribution with parameters $\alpha = 1$ and $\beta = 1$. After inference in a scene, the prior over $\lambda_c$ evolves by computing the inferred number of hallucinations at time $t$ (based on the difference between world state $W_t$ and observations $O_t$), updating $\alpha$ and $\beta$ as the conjugate prior over the Poisson distribution in the metacognition.

Beliefs about the miss rates evolve in an analogous manner. When $t = 0$, the detection rate $p_c$ for each category is initialized from a Beta distribution with parameters $\alpha = 1$ and $\beta = 1$ (equivalent to a uniform distribution). The parameters from the Beta distribution are then updated as the conjugate prior over the Geometric distribution, based on the inferred missed detections (by comparing $W_t$ against $O_t$).

## 3.2 INFERENCE

Given a collection of observations $\vec{O} = \{O_t\}_{t=1}^T$ (i.e., multiple sets of percepts from multiple world states) and the corresponding camera trajectories $\vec{cj}$, our goal is to infer $Pr(\vec{V}, \vec{W} | \vec{O}, \vec{cj})$, given by

$$Pr\left(\vec{V}, \vec{W} \mid \vec{O}, \vec{cj}\right) \propto \prod_{t=1}^T Pr\left(O_t | W_t, V_t, cj_t\right) Pr(V_t) Pr(W_t) \tag{1}$$

The posterior, Eq. 1, is approximated via Sequential Monte-Carlo using a particle filter. An estimate of the joint posterior can be sequentially approximated via:

$$Pr\left(\vec{V}, \vec{W} | \vec{O}, \vec{cj}\right) \approx Pr\left(\hat{V}_0^{\,0}\right) \prod_{t=1}^T Pr\left(O_t | \hat{V}_t^{\,t}, \hat{W}_t^{\,t}, cj_t\right) Pr\left(\hat{W}_t^{\,t}\right) Pr\left(\hat{V}_t^{\,t} | \hat{W}_{t-1}^{\,t}, \hat{O}_{t-1}^{\,t}\right)$$

$$\tag{2}$$

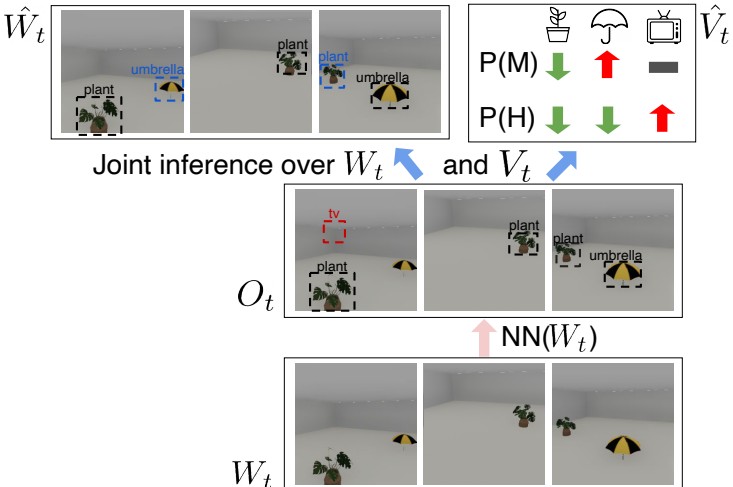

Figure 2: Conceptual figure depicting inference.

where $\hat{W}_1^T, \ldots, \hat{W}_T^T$ is the estimate $W_1, \ldots, W_T$ after $T$ observations, and $\hat{V}_1^T, \ldots, \hat{V}_T^T$ is the estimate $V_1, \ldots, V_T$ after $T$ observations. Here the transition kernel, $Pr(\hat{V}_t^t | \hat{W_{t-1}}^t, \hat{O_{t-1}}^t)$ is governed by the belief dynamics described in section 3.1.4.

### 3.2.1 INFERENCE PROCEDURE DETAILS

We sequentially approximate the joint posterior given in eq. 2 using a particle filter with 100 particles. The solution space to the inference problem that we consider is sparse, and Sequential Monte-Carlo methods can suffer from degeneracy and loss of diversity in these situations. Our inference approach solves these problems by implementing particle rejuvenation over objects, locations, and beliefs about $V$. Rejuvenation is conducted using a series of Metropolis-Hastings moves with data-driven proposals designed to obtain samples from $Pr(\vec{V}, \vec{W} | \vec{O}, \vec{cj})$.

During object rejuvenation, a new state $W_t'$ is proposed by either adding or removing an object from the current state $W_t$. Objects in $W_t$ have an equal chance of being removed to produce $W_t'$, and new objects are added based on a data-driven distribution. This distribution samples object categories from a categorical distribution biased towards categories observed in the current scene $t$ (see Appendix). With probability 0.5, location is sampled from a 3-D uniform distribution, and otherwise sampled from a data-driven function with location biased toward 3D points likely to have caused the 2D detections (see Appendix). The proposal for adding a new object or removing and existing object from the world state is accepted or rejected according to the MH algorithm.

After a proposed change to $W$, a second rejuvenation step is performed on locations, wherein an object in $W$ is randomly selected (with equal probability) to have a new location proposed. With probability 0.5, the new location is drawn from a multivariate normal distribution centered on the previous location, and it is otherwise sampled from a data-driven proposal (see Appendix). The proposed world-state $W'$ with a perturbed location is then accepted or rejected according to the MH algorithm.

Finally, a new metacognition $V'$ is proposed by perturbing $V$. Each parameter in the $|C|$-by-2 matrix is perturbed, with new values generated for $V(i,j)_t$ and for $V(i,j)_{t-1} \forall i,j$. The new value for each $V(i,j)_t$ is sampled from the appropriate distribution (Beta for misses, Gamma for hallucinations) with $\alpha_t$ and $\beta_t$, while the new value for $V(i,j)_{t-1}$ is sampled with $\alpha_{t-1}$ and $\beta_{t-1}$. These new values are accepted or rejected according to the MH algorithm.

This three-step rejuvenation process for world states, locations, and metacognition is done for each particle for 500 iterations and the last state reached in the chain is used as the new rejuvenated particle.

### 3.2.2 ESTIMATING $V$

After all $T$ time steps in the particle filter, we estimate $V_T$ by taking the expectation of the marginal distribution by averaging across particles weighted by their likelihood $l$: $\hat{V}_{T,\mu}^T = E[\hat{V}_T^T|\vec{O}] = \frac{1}{M}\sum_{m=1}^M(\hat{V}_{T,m}^T * l_m)$, where $m$ indexes the particles. This $\hat{V}_{T,\mu}^T$ is the final estimate of the belief about the true $V$ after all observations have been observed. Given $\hat{V}_{T,\mu}^T$, the posterior predictive distribution is defined as:

$$Pr\left(\hat{\vec{W}}|\vec{O},\vec{cj},\vec{V}=\hat{V}_{T,\mu}^T\right) \propto \prod_{t=1}^T Pr\left(O_t|\hat{W}_t,cj_t,V_t=\hat{V}_{T,\mu}^T\right)Pr\left(\hat{W}_t\right) \qquad (3)$$

This posterior predictive distribution can then be used to make better inferences about world states for novel scenes, $W_{T+1},\ldots$, or for reassessing previous world states $W_1,\ldots,W_T$, that were originally inferring using a less informed $V$.

## 4 EXPERIMENTS

To evaluate METAGEN, we created a dataset of trajectories navigating simulated indoor scenes, sampling several viewpoints per trajectory. We ran several object detection models on these images and tested METAGEN's ability to infer the underlying world states and properties of the object detection models causing the detections.

### 4.1 DATASET

We evaluate METAGEN on a dataset of simulated indoor scenes observed from multiple perspectives by a moving agent (Figure 2). We generate scenes and render frames in the ThreeDWorld physical simulation platform (Gan et al., 2020). Each scene is initialized as an empty 13-unit-by-13-unit room. The number of objects in the scene is uniformly drawn from the counting numbers up to 4 and each object is uniformly assigned one of 5 object categories: potted plant, chair, bowl, tv, or umbrella. We chose a canonical object model for each category. We sequentially place objects at locations drawn from a uniform distribution, resampling if two objects are within 3 units of each other or the walls so as to avoid collisions.

The camera trajectory consists of camera positions and focal points. Roughly, the camera circles the room looking toward the center, with noise from Gaussian processes added to the camera position and focal points. The details and parameters are discussed in the Appendix. We generate frames by querying the camera for an image at 20 linearly-spaced times.

We generated 100 videos in total, randomly split into two sets of 50 videos. The first set was used to test METAGEN's performance during online learning (Training), and the second set was used to test METAGEN's performance on a novel set of scenes after halting inference over $V$ (Test).

METAGEN's performance during online learning could be affected by the order in which the videos are processed (for instance, METAGEN might benefit if the most challenging videos appear later in the sequence). Our results therefore average METAGEN's performance across four different orderings of the videos. Videos were first randomly labeled from 1 to 50, and the four counterbalanced orders were: {1 ... 50}, {50 ... 1}, {26 ... 50, 1 ... 25}, and {25 ... 1, 26 ... 50}.

### 4.2 OBJECT DETECTION MODELS

To test METAGEN's capacity to learn and use a metacognition, we tested its performance using three modern neural networks for object detection as its perception module. The three networks were selected to represent a variety of state-of-the-art architectures. In particular, we use RetinaNet, a one-stage detector (Lin et al., 2017); Faster R-CNN, a two-stage detector (Ren et al., 2015), and DETR, a vision transformer (Carion et al., 2020). We process the outputs of the object detectors by filtering for the 5 object categories present in the scenes and performed Non-Maximum Suppression (NMS) with an IoU (Intersection over Union) threshold of 0.4 (applied only for RetinaNet and

Faster R-CNN, as it is not typically used for DETR). As input to METAGEN, we took the top five highest-confidence detections per frame.

## 4.3 METRICS USED

We evaluate METAGEN's performance in two ways: 1) by testing whether it can learn an accurate metacognition of the object detection module and 2) by testing whether the learned metacognition leads to improved accuracy.

To measure how well METAGEN learned a metacognition, we calculated the mean squared error (MSE) between the inferred visual system $\hat{V}$ and the true $V$. The true $V$ was calculated for each object detection system. The MSE is given by

$$\text{MSE} = \frac{1}{2|C|} \sum_{c \in C} \left( (\lambda_c - \hat{\lambda}_c)^2 + (p_c - \hat{p}_c)^2 \right) \tag{4}$$

where $C$ is the the set of object classes.

To measure METAGEN's object detection accuracy, we could not use the standard mean Average Precision (mAP) metric because the outputs of METAGEN are points in 3D space rather than 2D bounding boxes. But we still needed a metric by which to compare METAGEN's object detection accuracy to those of the neural networks without a metacognition.

To solve this problem, we projected METAGEN's inferences about the 3D location of objects to a point on each 2D image, and for the neural networks, we took the centroid of the bounding boxes. This way, METAGEN and the neural networks' detections have the same format: an object label and a point on the image.

To assess object detection accuracy on a given frame, we took the Jaccard similarity coefficient (the Intersection over Union for sets) of the object labels that were detected and the objects that were actually present in a give frame. The Jaccard similarity is given by:

$$J(\mathbb{D}, \mathbb{G}) = \frac{|\mathbb{D} \cap \mathbb{G}|}{|\mathbb{D} \cup \mathbb{G}|} \tag{5}$$

where $\mathbb{D}$ is the set of object labels that were detected (i.e. $\mathbb{D} = \{$chair, chair, bowl$\}$) and $\mathbb{G}$ is the set of objects in the ground-truth world state (i.e. $\mathbb{G} = \{$chair, bowl$\}$). (In this example, $J(\mathbb{D}, \mathbb{G}) = \frac{2}{3}$.)

A limitation of this metric is that it does not directly assess the spatial accuracy of the detections: only the categorical accuracy per frame. That said, highly inaccurate inferences about a 3D location would result in 2D projections onto the wrong frames, so this metric indirectly captures spatial accuracy. Averaging across frames yields an accuracy per video, and averaging across videos yields an overall accuracy.

## 5 COMPARISON MODELS

Throughout, we test METAGEN's performance while it is learning a metacognition as it processes videos (METAGEN Online Learning), and its performance when learning is halted and all inferences about world states are conditioned on the learned metacognition $\hat{V}$ (METAGEN Learned $\hat{V}$). We test METAGEN Online Learning on our Training set, and then test METAGEN Learned $\hat{V}$ on both the Training set (assessing if its accuracy changes when it can retrospectively re-evaluate its inferences after learning a metacognition), and the Test set.

To test if the learned metacognition confers a benefit beyond what can be obtained from the neural networks alone, we used two comparison models. The first comparison model (NN Output / META-GEN Input) consists of the processed neural network output, which served as an input to METAGEN (Section 4.2).

Our second comparison model (NN Output with Fitted Threshold) served as a more stringent test that compared METAGEN against the Neural Network output, fit to maximize accuracy. For each neural network, we fit the confidence threshold so as to maximize accuracy on the first 50 videos using a grid-search. This required access to the ground-truth object labels, giving an advantage to

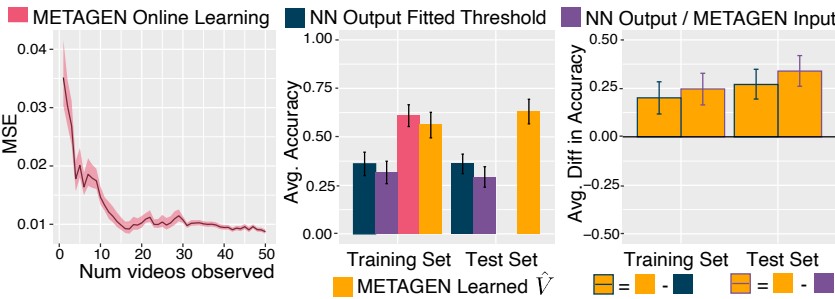

Figure 3: Results for METAGEN and comparison models with DETR. (A) METAGEN Online Learning MSE as a function of scenes observed. (B) Average accuracy for all models in Training Set and Test Set. (C) Accuracy difference between METAGEN Learned $\hat{V}$ and the two baseline models. Positive values indicated METAGEN outperforms the comparison models.

this comparison model over METAGEN. The values of the fitted threshold were 0.20 for DETR, 0.14 for RetinaNet, and 0.00 for Faster R-CNN.

## 6    RESULTS

To assess METAGEN's ability to learn an accurate metacognitive representation $V$, we examined the MSE as a function of the number of observed videos. Figure 3A shows the dynamics of the MSE for $V$ for the DETR model. 95% CIs over the MSE were calculated for each of the four runs by bootstrapping over the 100 particles. After observing just 10 videos, the MSE dropped from 0.035 to 0.014, and after training videos, the MSE was at 0.009. Results were similar for RetinaNet (final MSE at 0.023) and Faster R-CNN (final MSE at 0.010). This demonstrates the speed with which METAGEN is able to learn an accurate metacognition $V$, without access to ground-truth world states.

Does the learned metacognition enable METAGEN to make better inferences about what's in a scene? Figure 3B shows a barplot depicting the accuracies of all four models (Section 5) using DETR. We find that METAGEN outperforms the neural networks even with the fitted confidence threshold. The error bars represent 95% confidence intervals calculated by bootstrapping over the videos. METAGEN outperformed the two comparison models for all three object detection models (Table 1).

To evaluate the magnitude of the difference between METAGEN with Learned $\hat{V}$ and the two Neural Network comparison models (NN Output / METAGEN input and NN Output with Fitted Threshold), we calculated average accuracy difference in accuracy and boostrapped the mean difference to obtain a 95% confidence intervals over performance difference. METAGEN showed a substantial improvement against both comparison models (Figure 3C for DETR; Table 2 for full results).

Finally, to confirm that METAGEN's success can be attributed to its learned metacognition, rather than purely to the Spelke object principles, we tested METAGEN with an erroneous metacognition consisting of low ($1e-12$) miss rates and high ($1.0$) hallucination rates. Under these parameters, METAGEN can only distinguish real detections from hallucinations based on whether the pattern of detections across frames adheres to Spelke principles. This lesioned METAGEN led to a degenerate solution where all detections were always treated as hallucinations, for all object detection models, with an accuracy of $0.228$, confirming that Spelke principles alone cannot account for METAGEN's success.

## 7    DISCUSSION

We proposed that metacognition—a metarepresentation of one's own computational processes— can be a powerful tool for increasing a system's robustness, enabling it to decide when to accept or reject proposed representations obtained from pre-trained systems. We presented an implementation

Table 1: Main results

| Perceptual Module | Model | Version | Avg. Accuracy Training | Avg. Accuracy Test |
|---|---|---|---|---|
| **DETR** | NN Output | Fitted threshold | 0.361 (0.300, 0.422) | 0.361 (0.313, 0.412) |
| | | METAGEN input | 0.315 (0.257, 0.373) | 0.292 (0.241, 0.346) |
| | METAGEN | Online learning | **0.609** (0.552, 0.667) | - |
| | | Learned $\hat{V}$ | 0.562 (0.494, 0.627) | **0.631** (0.569, 0.695) |
| **RetinaNet** | NN Output | Fitted threshold | 0.582 (0.512, 0.650) | 0.543 (0.462, 0.568) |
| | | METAGEN input | 0.510 (0.452, 0.568) | 0.539 (0.468, 0.609) |
| | METAGEN | Online learning | **0.671** (0.595, 0.744) | - |
| | | Learned $\hat{V}$ | 0.666 (0.594, 0.735) | **0.647** (0.562, 0.730) |
| **FasterR-CNN** | NN Output | Fitted threshold | 0.656 (0.589, 0.724) | 0.630 (0.554, 0.705) |
| | | METAGEN input | 0.656 (0.589, 0.724) | 0.630 (0.554, 0.705) |
| | METAGEN | Online learning | **0.711** (0.639, 0.775) | - |
| | | Learned $\hat{V}$ | 0.697 (0.628, 0.762) | **0.650** (0.563, 0.732) |

Table 2: Accuracy difference between METAGEN learned $V$ and NN output with fitted threshold

| Perceptual Module | Avg. Difference on Test Set | % Increase in Accuracy |
|---|---|---|
| **DETR** | **0.270** (0.195, 0.350) | **74.8**% |
| **RetinaNet** | **0.104** (0.0782, 0.130) | **19.2**% |
| **Faster R-CNN** | **0.020** (0.000, 0.004) | **3.1**% |

of this idea, METAGEN, applied to the context of inferring what objects were in a scene. Given a set of observations by an object detection model with unknown performance, METAGEN performed joint inference over a metacognitive representation of the system and over the objects causing the detections. We tested METAGEN using a variety of modern object detection models, and found that METAGEN can quickly learn an accurate metacognition and use it to correct errors from the classification system, improving the system's overall accuracy.

That improvement in accuracy was observed on a custom dataset of synthetic images, in some ways quite different from those the object detection systems were trained on. Thus, our results show that METAGEN can be particularly helpful in improving the ability of an object detection system to generalize to out-of-sample data, especially when considering that METAGEN does not require any internal access to the model or knowledge of its performance metrics. By learning a metacognition for how the detection system performs on a new dataset, METAGEN is able filter the outputs of the detection system to be tailored to a new, out-of-sample data.

While our focus was on object detection, we believe that similar ideas can be applied to related domains. For instance, face-detection models could be augmented with a metacognition that learns to represent its own accuracy, conditioned on an expectation that all faces should be, in principle, equally discriminable. This could allow a METAGEN model to learn its biases (e.g., such as learning that it has poor performance for faces from people of color). From this standpoint, METAGEN can also support explainable AI by generating a simplified metarepresentations of a model's performance, which can be easily analyzed. Similarly, because METAGEN can help infer the presence of missed objects, or flag hallucinations in an image, this approach could be fruitful as a way of generating additional training data for self-supervised learning.

Finally, our work highlights how computational solutions that appear in humans can be fruitful for approaching related problems in machine learning and AI. In recent years, models that capture human reasoning have received substantial attention (e.g., Smith et al., 2019; Baker et al., 2017; Jara-Ettinger et al., 2019; Battaglia et al., 2013). By understanding how humans leverage representations of their cognitive processes to create more nuanced and accurate representations of the world, we may also be better able to design human-like artificial cognition.

REPRODUCIBILITY

To ensure reproducibility, we provide all of the code for the METAGEN model, producing the dataset, and the analyses conducted. Please see the anonymous git repo linked below for the code and for a demo:

https://anonymous.4open.science/r/MetaGen-0391/

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

# A    APPENDIX

The METAGEN model and inference were in implemented in the Julia-based probablistic programming language Gen (Cusumano-Towner et al., 2019).

## A.1    ADDITIONAL GENERATIVE MODEL PARAMETERS

Implementation of Spelke principles was as follows. World states without object persistence were not included in $\mathbb{W}$, which is equivalent to implicitly setting their prior probability to 0. The assumption that two objects cannot occupy the same region in space was implemented through a prior over $\mathbb{W}$ where the probability of having one object near another decreased according to a Gaussian distribution with $\sigma^2 = 1$.

To account for noise in a network's location detection $O_t$, each detection was modeled as having 2D spatial noise, following a Gaussian with $\sigma_{x,y} = 40$ pixels.

Finally, the full generative model requires specifying a prior distribution over camera positions and focal points (although these are observable), set as uniform over 3D space, and a prior distribution over expected number of objects in a scene, sampled from a geometric distribution with parameter $p = 0.9$ (with a uniform prior over category type).

## A.2    DETAILS ABOUT THE INFERENCE PROCEDURE

During rejuvenation, new objects are proposed to be added to $W_t'$ using a data-driven distribution. The category of the object is comes from a categorical distribution, where 10% of the weight is divided evenly among the categories, and the other 90% is divided proportionally to the number of times that object was observed in the scene $t$. With probability 0.5, the location is sampled from a 3-D uniform distribution, and otherwise sampled from a data-driven function. Using the data-driven function, the point is sampled based on proximity to the line-segment that, when projected onto the 2D image, would result in the point where the detection was observed. The probability of proposing a particular point decreases with the distance from this line segment, following a Gaussian with $\sigma^2 = 0.01$.

In the second rejuvenation step a new location is proposed for an object. With probability 0.5, the new location is drawn from a multivariate normal distribution centered on the previous location, with $\sigma_{x,y,z} = 0.01$. Otherwise, it is sampled as described in the previous paragraph (based on proximity to the line segment that would results in the detection's 2D location).

## A.3    DETAILS ABOUT THE CAMERA TRAJECTORY IN THE DATASET

In our dataset of simulated indoor scenes, the agent's trajectory consists of a series of camera positions and focal points. The camera trajectory is a circular path around the periphery of the room with noise generated by a Gaussian process with an RBF kernel with $\sigma = 0.7$ and $\ell = 2.5$. The height of the camera is held constant at $y = 2$. The focal point trajectory is a Gaussian process with a mean above the center of the floor and component-wise parameters $\sigma = 0.7$ and $\ell = 2$. Figure 4 shows some example trajectories.

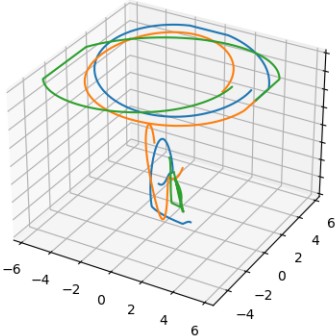

Figure 4: Three sampled agent trajectories. The wide circular patterns at the top are camera positions, and the smaller patterns near the bottom are camera focal points.

