# OpenReview forum: "Learning a metacognition for object detection"
_ICLR.cc/2022/Conference — ICLR 2022 Submitted_

### Official Review · Reviewer_1uYX · 2021-11-02

**Correctness:** 2
**Technical Novelty And Significance:** 3
**Empirical Novelty And Significance:** 3
**Recommendation:** 5
**Confidence:** 4

**Main Review:**

In the new datasets collected by the author, the model achieves significant improvement based on the existing detection algorithms. In addition, the idea of using METAGAN to deal with object detection proposes an entire new perspective to address the detection problem, which is very interesting to me.

My main concern to accept this paper is that the experiments are not convincing. The problem setting in this paper is quite different with the other papers for object detection, and the dataset for evaluation is more like a toy dataset, with simple contexts and limited images for training. For object detection, the complicated context is one of the main challenges, and it is unfair to compare models with detectors for 2D images since they are not designed to capture the trajectory between each pair if the frame. I wonder whether the author can report the results of the proposed module based on the large-scale public video detection datasets (e.g., ImageNet VID [1]) , and compare the results with the detectors for video detection [2] [3], where the models can also capture the relationship between different frames. It will make the proposed framework far more convincing.

[1] Russakovsky O, Deng J, Su H, et al. Imagenet large scale visual recognition challenge[J]. International journal of computer vision, 2015, 115(3): 211-252.

[2] Han M, Wang Y, Chang X, et al. Mining inter-video proposal relations for video object detection[C]//European Conference on Computer Vision. Springer, Cham, 2020: 431-446.

[3] Gong T, Chen K, Wang X, et al. Temporal ROI Align for Video Object Recognition[C]//Proceedings of the AAAI Conference on Artificial Intelligence. 2021, 35(2): 1442-1450.



**Summary Of The Paper:**

This paper proposes a new METAGAN module which captures the relationship between objects and detections in each time slot, and it refines the detection results by eliminating the false predictions.


**Summary Of The Review:**

Overall, my point is on the fence. As the limitation I mention above, my pre-rebuttal score is slightly below the acceptance bar. If the author can validate the effectiveness of the proposed method on the challenging public benchmarks, I'd like to change the score.

---

### Official Review · Reviewer_Rg2j · 2021-11-02

**Correctness:** 2
**Technical Novelty And Significance:** 2
**Empirical Novelty And Significance:** 4
**Recommendation:** 6
**Confidence:** 4

**Main Review:**

The topic of this article is a very relevant one: How to account for an AI inaccuracy for decision processes? The proposed idea of a metacognitive component is a well inspired one. The proposed model is sound and well described, but simple: object detection algorithms do not typically have flat probabilities of hallucination or miss, but tend to be more likely to miss objects that are further away and viewed under specific angles. In a realistic scenario, specific instances of an object class may be more prone to misses. The proposed model does not account for that.

Another issue with the article is the very synthetic nature of the experimental setup. The history of computer vision tells us that results showed on artificial data rarely transfer to real-world situations. In particular, as the authors note, the chosen scenario is quite far from the data the detectors were trained on and the performance reported seems quite low for such a simple problem. Some form of domain adaptation or fine-tuning may be required here to compensate for this and make the baselines more realistic.

Although it is a good idea to use a simplified scenario for experimental purposes, I have some concern that the performance improvement may only be due to the artificially low performance of the baseline algorithms on this task. The fact that the error metric is non-standard and does not account for localisation makes it more difficult to really assess how meaningful are the results.

Additionally, it is unclear from the experimental results that the metacognitive model could be generalised easily to other scenarios due to the low variety in conditions between the training and test data.


**Summary Of The Paper:**

This article proposes a metacognition model for object detection. In this article, metacognition is understood as the ability for a system to monitor its own beliefs and account for its own accuracy. The proposed model learns the probability for a detector to hallucinate (false positive) or fail to detect an object. To this end, the model assumes some prior world knowledge drawn from Psychology: the Spelke principles: Object persistence, continuous motion and non-colocation of objects.

The proposed model is trained and tested on a synthetic dataset and the experimental results show an improvement in detection performance compared to standard models.

**Summary Of The Review:**


In summary, this is an interesting research topic, but the significance of the article is limited by the rather simple model choices and experimental setup. Because of those choices, it is unclear that the proposed method would generalise to other setups.

---

### Official Review · Reviewer_RfQP · 2021-11-03

**Correctness:** 3
**Technical Novelty And Significance:** 2
**Empirical Novelty And Significance:** 2
**Recommendation:** 5
**Confidence:** 3

**Main Review:**

# Strengths

1. I think the motivation behind the paper is strong and is a good fit for ICLR. The idea of being able to take a noisy object detector and learn how to denoise its outputs using a set of sensible priors, without retraining or needing to know the inner workings of the object detector itself, is a simple and attractive one to me. The paper does a good job of explaining this motivation and framing the work.
2. The METAGEN model proposed in Section 3 is clearly explained and seems like a reasonable approach to impose the various desirable metacognitive priors.
3. The experiment uses a range of object detectors to show that the proposed METAGEN model can provide benefit to any generic noisy black-box detector.
4. The work appears to be fully reproducible via code at https://anonymous.4open.science/r/MetaGen-0391/

# Weaknesses

1. What motivated the choice of dataset? Given the simplicity of generating data, why not make more complex scenes, and more complex trajectories? For example, are there any object-to-object occlusions in the dataset? Based on the videos and trajectories provided, it seems possibly not? These are interesting because they can easily confuse detectors and are a good motivation for trying to introduce notions of object permanence into models. Does the model hold up in more cluttered environments where real occlusions occur? The approach would be, I think, considerably more compelling if it was shown to work on more than one dataset, particularly without significant tuning of METAGEN hyperparameters between the two.

2. How were the initial three models trained? I wonder if the main difference in improvement by applying METAGEN is more to do with how strong those models were to begin with. It might be interesting to show e.g. what is the relative improvement if all three models are pre-trained to the same level of test performance? It might also be interesting to see what happens as the detection models are given different amounts of pre-training. How does the performance vs training data size curve look for the models with and without METAGAN?

3. The robotics community has been using methods like particle filtering for quite a long time to deal with issues of unreliable sensing / perception. Although they do not frame their work in the same metacognitive light, the end effect is somewhat similar - the perception model provides noisy frame-by-frame inputs while a dynamics/state space model is used to smooth the estimates enforcing notions like object permanence. I'm blanking on a good reference that's closest to this work, but it would be nice to see some effort to make a connection here.

4. There are certain experimental details which I think could be added to improve understanding.  For example, the order of training scene samples clearly has an effect, but how critical is it? What are the effects of change non-maximum suppression / not filtering to top five highest-confidence detections per frame? How sensitive are the results to the various hyperparameters involved in the SMC optimization?
Were there any observed or hypothetical failure modes of this metacognitive approach, where it might actually worsen the outcome? One can imagine it could be vulnerable to certain kinds of cascading error, for example.

5. Sec 5 - for the second comparison model, is this thresholding the softmax classification output of the detector? If so, is the 0.00 fitted threshold value for Faster R-CNN rather surprising? Is it close to being perfect in detection performance?

6. I’m unclear why object label-level Jaccard similarity was chosen as the metric. I understand that it would indirectly capture some notion of spatial accuracy, but could this be done more explicitly? e.g. by also measuring whether projected objects or predicted bounding box centroids lie within the ground truth bounding box?

# Minor points

1. Fig 2 caption chould ideally be self-explanatory without referring to text. Why is plant in blue in top panel?

2. The two different colors in Fig 3C is hard to see - maybe use dashed outlines to make it more obvious?

3. Table 2 Faster R-CNN row has a numerical typo

4. It would be good to see Fig 3 for the other two models for comparison, perhaps in the Appendix if they are not particularly interesting?

5. “the assumption of spatio-temporal continuity implies that any moving object must exhibit a continuous trajectory… we do not implement this constraint, although doing so is a solved problem” -- I think calling this “solved” may be a bit reductive; could this statement be qualified a little more?

6. Misc typos; “removing and existing” -> an; “were originally inferring” -> inferred

7. The bounds in Tables 1 and 2 could be simplified to a single +/- number; also probably could be shown to 2 decimal places to improve readability.

**Summary Of The Paper:**

This paper proposes a metacognitive framework to augment object detection models to help improve them without human input.
The framework is based on learning to detect and fix a model’s output if it violates early object constraints (Spelke principles), namely object permanence, cohesion, and spatiotemporal continuity.
The framework, called METAGEN, does not require access to the internal state of the detection model but operates only on its outputs, maintaining a metacognitive representation which summarizes the model’s performance (propensity to hallucinate or incorrectly miss objects of different classes).
The representation is initialized for a given scene from a learned and evolving prior, and is updated along with the estimated world state using a particle filter with various rejuvenation steps over objects, locations and metacognitive parameters.
METAGEN is tested using single-stage (RetineNet), two-stage (FasterR-CNN), and transformer (DETR) models of object detection within a dataset gathered using the synthetic ThreeDWorld environment, and is shown to improve test performance both during inference and after stopping inference on both training and new test scenes.


**Summary Of The Review:**

I think this is an interesting paper that could merit publication but I am concerned that the experimental side is lacking for the reasons mentioned above. I would say it is below the threshold for publication as I read it right now, but I look forward to hear clarifications from the authors and will take into consideration the Q&A with all the reviewers.

---

### Official Review · Reviewer_RHiF · 2021-11-03

**Correctness:** 3
**Technical Novelty And Significance:** 2
**Empirical Novelty And Significance:** Not applicable
**Recommendation:** 3
**Confidence:** 3

**Main Review:**

$\textbf{Strengths}$

The paper introduces an interesting formulation on how to incorporate various consistency constraints in video object detection. The proposed METAGEN framework is simple and intuitive, and can easily be extended to other applications.

$\textbf{Weaknesses}$
1. I am not sure if the formulation used in the paper can be treated as a traditional object detection problem. The authors use a video and take into account the camera trajectory, which is additional information unavailable to the baseline 2D detection method. Therefore directly comparing to 2D object detection approaches seems unfair. Additionally, the use of a video makes it closely aligned to object tracking or video object detection, where the ideas of enforcing cohesions and spatiotemporal continuity have previously been explored. (For example [a], [b], and [c]). The authors should discuss these (and possibly other) methods in their related work, and highlight how their proposed approach is different. Additionally, comparisons to these methods that utilize video/camera information would be a better baseline.

2. The experiments are only shown on a single synthetic dataset, which is extremely small in size (only 100 videos). Additionally, the diversity in objects in limited (5 objects), and the generated scenes don't account for background noises (which is what you would get in a real world setting).

3. Qualitative results showing how the proposed approach improves on the baseline detections is missing. Even though there is an improvement in the numbers, its hard to understand how these improvements translate visually.

4. Evaluating the detection accuracy using only centroids makes it hard to correctly understand the improvements in detection performance as it gives no sense of the bounding box area. As you already assume you know the camera position, it should be possible to map the 3D bounding box to a 2D box. Additionally, as mentioned in the paper, the accuracy metric computed doesn't take into account the spatial correctness. Is there a reason why the authors did not compute the L1 distances between the projected 1D bounding box point and ground truth?

5. I am confused how the true $V$ is obtained to compute the MSE given in Equation (4). It would be great if the authors could clarify this.

[a] Zhu, Xizhou, et al. "Flow-guided feature aggregation for video object detection." Proceedings of the IEEE International Conference on Computer Vision. 2017.

[b] Bertasius, Gedas, Lorenzo Torresani, and Jianbo Shi. "Object detection in video with spatiotemporal sampling networks." Proceedings of the European Conference on Computer Vision (ECCV). 2018.

[c] Wang, Qiang, et al. "Fast online object tracking and segmentation: A unifying approach." Proceedings of the IEEE/CVF Conference on Computer Vision and Pattern Recognition. 2019.



**Summary Of The Paper:**

The paper proposes a new method to incorporate Spelke's principals of object perception as constraints to improve the performance of an out-of-the-box object detector. This is done via defining a hierarchical generative model which defines "metacognitive" priors over the a set of observations. Through joint inference over these metacognitive priors and new unobserved states, the method outputs better object detections. The authors show improved performance on a synthetic dataset which contains scenes rendered in a virtual environment.

**Summary Of The Review:**

The paper presents an interesting way of enforcing consistency constraints in object detection. However, I feel that the experimental evaluation and overall novelty when compared to existing approaches in video object detection/tracking is lacking. Therefore, I feel that this paper, in its current state, is not ready for publication.

---

### Author Response · Authors · 2021-11-18
**Thank you to our reviewers**

Dear Reviews,

Thank you for your time and feedback! We have learned a lot from your responses. The purpose of this letter is not to try to change the scores (as it was infeasible for us to address the reviewers’ concerns within the rebuttal period timeline), but rather to thank the reviewers and answer some questions that had arisen.

One common thread across reviews were concerns about our dataset. Our model requires camera positions and stationary objects, which limits the dataset we could possibly use. To our knowledge, no dataset of videos of objects from a moving camera with known positions exist. (This is why we could not use a standard dataset like ImageNet VID.) If you know of such a dataset, we would love to hear it! This is what motivated our custom dataset. We are in the process of further improvements to our custom datasets (like having more interesting / realistic backgrounds) that might make it seem more reasonable.

Another common thread across reviews was that our accuracy metric should directly account for location. We plan to improve this metric in future versions.

We think these reviews pointed to an issue in how we framed our model: we don’t necessarily want to frame it as improving existing object-detection systems, since we can’t show that on existing, standard datasets. What we do want to claim is that, from camera information and detections, it is possible to learn a metacognition for a visual system in a particular environment, and that having that metacognition can help to infer what’s really there.

Thank you again for reviewing our paper. We really appreciate your time and the thought that you put into the reviews. Future versions of this project will take this feedback into account.

Gratefully,
Anonymous author

---

> ### Comment · Reviewer_RfQP · 2021-11-30
> **Thank you to the authors**
>
> I just want to say thank you to the authors for posting this comment and acknowledging the efforts of the reviewers. It is much appreciated and I will look forward to see if I can spot their newly improved manuscript when it is published in the near future.

---

> ### Comment · Reviewer_1uYX · 2021-11-30
> **Look forward to the refined version in the future**
>
> Thx for the reply of the author. This paper definitely proposes a very interesting paradigm for object detection, and also broaden my understanding. I look forward to the refined version in the future when the proposed concerns (e.g., datasets and evaluation) are resolved.

---

### Decision · Program_Chairs · 2022-01-20

**Decision:**

Reject

**Comment:**

This work investigates a metacognition model for object detection. Reviewer RHiF wrote the best summary for this work:

The paper proposes a new method to incorporate Spelke's principles of object perception as constraints to improve the performance of an out-of-the-box object detector. This is done via defining a hierarchical generative model which defines "metacognitive" priors over the a set of observations. Through joint inference over these metacognitive priors and new unobserved states, the method outputs better object detections. The authors show improved performance on a synthetic dataset which contains scenes rendered in a virtual environment.

All reviewers agree that this is a really novel and interesting approach of enforcing consistency constraints in object detection, but had various issues with the experiments. At its current state, I believe it would make a very strong workshop paper, but not read for the ICLR 2020 conference. The authors found the reviews to be helpful, in particular, advice about the dataset construction and metric definitions, and I believe that future versions of this work will be significantly improved. We look forward to reading a revised version of this work in a high impact journal or future ML conference, good luck!